# CGANS WITH PROJECTION DISCRIMINATOR

**Takeru Miyato[1], Masanori Koyama[2]**
miyato@preferred.jp
koyama.masanori@gmail.com
[1]Preferred Networks, Inc. [2]Ritsumeikan University

## ABSTRACT

We propose a novel, *projection based* way to incorporate the conditional information into the discriminator of GANs that respects the role of the conditional information in the underlining probabilistic model. This approach is in contrast with most frameworks of conditional GANs used in application today, which use the conditional information by concatenating the (embedded) conditional vector to the feature vectors. With this modification, we were able to significantly improve the quality of the class conditional image generation on ILSVRC2012 (ImageNet) 1000-class image dataset from the current *state-of-the-art* result, and we achieved this *with a single pair of a discriminator and a generator*. We were also able to extend the application to super-resolution and succeeded in producing highly discriminative super-resolution images. This new structure also enabled high quality *category transformation* based on parametric functional transformation of conditional batch normalization layers in the generator. The code with Chainer (Tokui et al., 2015), generated images and pretrained models are available at https://github.com/pfnet-research/sngan_projection.

## 1 INTRODUCTION

Generative Adversarial Networks (GANs) (Goodfellow et al., 2014) are a framework to construct a generative model that can mimic the target distribution, and in recent years it has given birth to arrays of state-of-the-art algorithms of generative models on image domain (Radford et al., 2016; Salimans et al., 2016; Ledig et al., 2017; Zhang et al., 2017; Reed et al., 2016). The most distinctive feature of GANs is the discriminator $D(x)$ that evaluates the divergence between the current generative distribution $p_G(x)$ and the target distribution $q(x)$ (Goodfellow et al., 2014; Nowozin et al., 2016; Arjovsky et al., 2017). The algorithm of GANs trains the generator model by iteratively training the discriminator and generator in turn, with the discriminator acting as an increasingly meticulous critic of the current generator.

Conditional GANs (cGANs) are a type of GANs that use conditional information (Mirza & Osindero, 2014) for the discriminator and generator, and they have been drawing attention as a promising tool for class conditional image generation (Odena et al., 2017), the generation of the images from text (Reed et al., 2016; Zhang et al., 2017), and image to image translation (Kim et al., 2017; Zhu et al., 2017). Unlike in standard GANs, the discriminator of cGANs discriminates between the generator distribution and the target distribution on the set of the pairs of generated samples $x$ and its intended conditional variable $y$. To the authors' knowledge, most frameworks of discriminators in cGANs at the time of writing feeds the pair the conditional information $y$ into the discriminator by naively concatenating (embedded) $y$ to the input or to the feature vector at some middle layer (Mirza & Osindero, 2014; Denton et al., 2015; Reed et al., 2016; Zhang et al., 2017; Perarnau et al., 2016; Saito et al., 2017; Dumoulin et al., 2017a; Sricharan et al., 2017). We would like to however, take into account the structure of the assumed conditional probabilistic models underlined by the structure of the discriminator, which is a function that measures the information theoretic distance between the generative distribution and the target distribution.

By construction, any assumption about the form of the distribution would act as a regularization on the choice of the discriminator. In this paper, we propose a specific form of the discriminator, a form motivated by a probabilistic model in which the distribution of the conditional variable $y$ given $x$ is

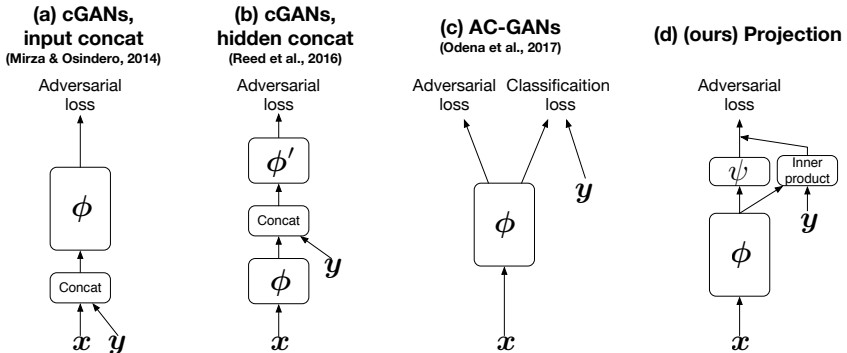

Figure 1: Discriminator models for conditional GANs

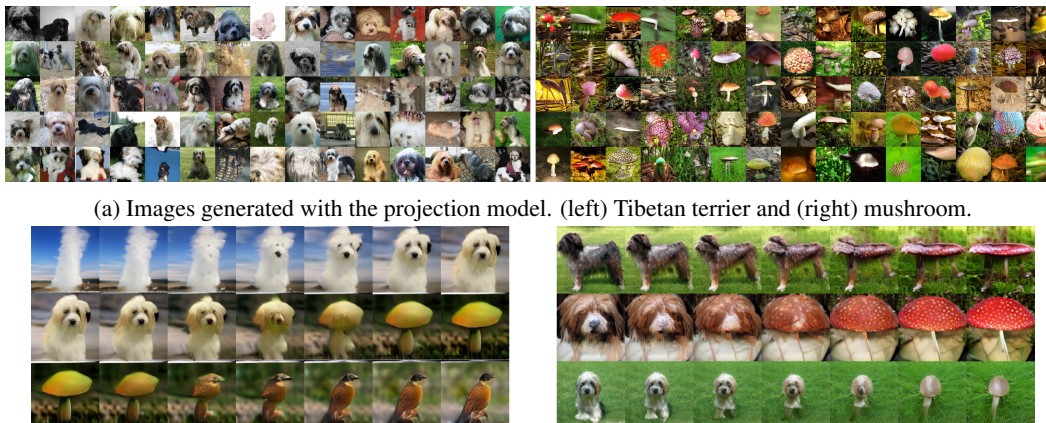

(a) Images generated with the projection model. (left) Tibetan terrier and (right) mushroom.

(b) (left) Consecutive category morphing with fixed $z$. geyser $\rightarrow$ Tibetan terrier $\rightarrow$ mushroom $\rightarrow$ robin. (right) category morphing from Tibetan terrier to mushroom with different value of fixed $z$

Figure 2: The generator trained with the *projection* model can generate diverse set of images. For more results, see the experiment section and the appendix section.

discrete or uni-modal continuous distributions. This model assumption is in fact common in many real world applications, including class-conditional image generation and super-resolution.

As we will explain in the next section, adhering to this assumption will give rise to a structure of the discriminator that requires us to take an inner product between the embedded condition vector $y$ and the feature vector (Figure 1d). With this modification, we were able to significantly improve the quality of the class conditional image generation on 1000-class ILSVRC2012 dataset (Russakovsky et al., 2015) with *a single pair of a discriminator and generator* (see the generated examples in Figure 2). Also, when we applied our model of cGANs to a super-resolution task, we were able to produce high quality super-resolution images that are more discriminative in terms of the accuracy of the label classifier than the cGANs based on concatenation, as well as the bilinear and the bicubic method.

## 2   THE ARCHITECTURE OF THE CGAN DISCRIMINATOR WITH A PROBABILISTIC MODEL ASSUMPTIONS

Let us denote the input vector by $x$ and the conditional information by $y$[1]. We also denote the cGAN discriminator by $D(x, y; \theta) := \mathcal{A}(f(x, y; \theta))$, where $f$ is a function of $x$ and $y$, $\theta$ is the parameters of $f$, and $\mathcal{A}$ is an activation function of the users' choice. Using $q$ and $p$ to designate the true distributions and the generator model respectively, the standard adversarial loss for the

---

[1]When $y$ is discrete label information, we can assume that it is encoded as a one-hot vector.

discriminator is given by:

$$\mathcal{L}(D) = -E_{q(\boldsymbol{y})}\left[E_{q(\boldsymbol{x}|\boldsymbol{y})}\left[\log(D(\boldsymbol{x}, \boldsymbol{y}))\right]\right] - E_{p(\boldsymbol{y})}\left[E_{p(\boldsymbol{x}|\boldsymbol{y})}\left[\log(1 - D(\boldsymbol{x}, \boldsymbol{y}))\right]\right], \quad (1)$$

with $\mathcal{A}$ in $D$ representing the sigmoid function. By construction, the nature of the 'critic' $D$ significantly affects the performance of $G$. A conventional way of feeding $\boldsymbol{y}$ to $D$ until now has been to concatenate the vector $\boldsymbol{y}$ to the feature vector $\boldsymbol{x}$, either at the input layer (Mirza & Osindero, 2014; Denton et al., 2015; Saito et al., 2017), or at some hidden layer (Reed et al., 2016; Zhang et al., 2017; Perarnau et al., 2016; Dumoulin et al., 2017a; Sricharan et al., 2017) (see Figure 1a and Figure 1b). We would like to propose an alternative to this approach by observing the form of the optimal solution (Goodfellow et al., 2014) for the loss function, Eq. (1), can be decomposed into the sum of two log likelihood ratios:

$$f^*(\boldsymbol{x}, \boldsymbol{y}) = \log \frac{q(\boldsymbol{x}|\boldsymbol{y})q(\boldsymbol{y})}{p(\boldsymbol{x}|\boldsymbol{y})p(\boldsymbol{y})} = \log \frac{q(\boldsymbol{y}|\boldsymbol{x})}{p(\boldsymbol{y}|\boldsymbol{x})} + \log \frac{q(\boldsymbol{x})}{p(\boldsymbol{x})} := r(\boldsymbol{y}|\boldsymbol{x}) + r(\boldsymbol{x}). \quad (2)$$

Now, we can model the log likelihood ratio $r(\boldsymbol{y}|\boldsymbol{x})$ and $r(\boldsymbol{x})$ by some parametric functions $f_1$ and $f_2$ respectively. If we make a standing assumption that $p(\boldsymbol{y}|\boldsymbol{x})$ and $q(\boldsymbol{y}|\boldsymbol{x})$ are simple distributions like those that are Gaussian or discrete log linear on the feature space, then, as we will show, the parametrization of the following form becomes natural:

$$f(\boldsymbol{x}, \boldsymbol{y}; \theta) := f_1(\boldsymbol{x}, \boldsymbol{y}; \theta) + f_2(\boldsymbol{x}; \theta) = \boldsymbol{y}^{\mathrm{T}} V \boldsymbol{\phi}(\boldsymbol{x}; \theta_\Phi) + \psi(\boldsymbol{\phi}(\boldsymbol{x}; \theta_\Phi); \theta_\Psi), \quad (3)$$

where $V$ is the embedding matrix of $\boldsymbol{y}$, $\boldsymbol{\phi}(\cdot, \theta_\Phi)$ is a vector output function of $\boldsymbol{x}$, and $\psi(\cdot, \theta_\Psi)$ is a scalar function of the same $\boldsymbol{\phi}(\boldsymbol{x}; \theta_\Phi)$ that appears in $f_1$ (see Figure 1d). The learned parameters $\theta = \{V, \theta_\Phi, \theta_\Psi\}$ are to be trained to optimize the adversarial loss. From this point on, we will refer to this model of the discriminator as *projection* for short. In the next section, we would like to elaborate on how we can arrive at this form.

## 3 MOTIVATION BEHIND THE *projection* DISCRIMINATOR

In this section, we will begin from specific, often recurring models and show that, with certain regularity assumption, we can write the optimal solution of the discriminator objective function in the form of (3). Let us first consider the a case of categorical variable. Assume that $y$ is a categorical variable taking a value in $\{1, \ldots, C\}$, which is often common for a class conditional image generation task. The most popular model for $p(y|\boldsymbol{x})$ is the following log linear model:

$$\log p(y = c|\boldsymbol{x}) := \boldsymbol{v}_c^{p\mathrm{T}} \boldsymbol{\phi}(\boldsymbol{x}) - \log Z(\boldsymbol{\phi}(\boldsymbol{x})), \quad (4)$$

where $Z(\boldsymbol{\phi}(\boldsymbol{x})) := \left(\sum_{j=1}^C \exp\left(\boldsymbol{v}_j^{p\mathrm{T}} \boldsymbol{\phi}(\boldsymbol{x})\right)\right)$ is the partition function, and $\boldsymbol{\phi} : \boldsymbol{x} \mapsto \mathbb{R}^{d^L}$ is the input to the final layer of the network model. Now, we assume that the target distribution $q$ can also be parametrized in this form, with the same choice of $\boldsymbol{\phi}$. This way, the log likelihood ratio would take the following form;

$$r(y|\boldsymbol{x}) = \log \frac{q(y = c|\boldsymbol{x})}{p(y = c|\boldsymbol{x})} = (\boldsymbol{v}_c^q - \boldsymbol{v}_c^p)^{\mathrm{T}} \boldsymbol{\phi}(\boldsymbol{x}) - (\log Z^q(\boldsymbol{\phi}(\boldsymbol{x})) - \log Z^p(\boldsymbol{\phi}(\boldsymbol{x}))). \quad (5)$$

If we make the values of $(\boldsymbol{v}_c^q, \boldsymbol{v}_c^p)$ implicit and put $\boldsymbol{v}_c := (\boldsymbol{v}_c^q - \boldsymbol{v}_c^p)$, we can write $f_1(\boldsymbol{x}, y = c) = \boldsymbol{v}_c^{\mathrm{T}} \boldsymbol{\phi}(\boldsymbol{x})$. Now, if we can put together the normalization constant $-(\log Z^q(\boldsymbol{\phi}(\boldsymbol{x})) - \log Z^p(\boldsymbol{\phi}(\boldsymbol{x})))$ and $r(\boldsymbol{x})$ into one expression $\psi(\boldsymbol{\phi}(\boldsymbol{x}))$, we can rewrite the equation above as

$$f(\boldsymbol{x}, \boldsymbol{y}) := \boldsymbol{y}^{\mathrm{T}} V \boldsymbol{\phi}(\boldsymbol{x}) + \psi(\boldsymbol{\phi}(\boldsymbol{x})). \quad (6)$$

by using $\boldsymbol{y}$ to denote a one-hot vector of the label $y$ and using $V$ to denote the matrix consisting of the row vectors $\boldsymbol{v}_c$. Most notably, this formulation introduces the label information via an inner product, as opposed to concatenation. The form (6) is indeed the form we proposed in (3).

We can also arrive at the form (3) for unimodal continuous distributions $p(\boldsymbol{y}|\boldsymbol{x})$ as well. Let $\boldsymbol{y} \in \mathbb{R}^d$ be a $d$-dimensional continuous variable, and let us assume that conditional $q(\boldsymbol{y}|\boldsymbol{x})$ and $p(\boldsymbol{y}|\boldsymbol{x})$ are both given by Gaussian distributions, so that $q(\boldsymbol{y}|\boldsymbol{x}) = \mathcal{N}(\boldsymbol{y}|\boldsymbol{\mu}_q(\boldsymbol{x}), \Lambda_q^{-1})$ and

$p(\boldsymbol{y}|\boldsymbol{x}) = \mathcal{N}(\boldsymbol{y}|\boldsymbol{\mu}_p(\boldsymbol{x}), \boldsymbol{\Lambda}_p^{-1})$ where $\boldsymbol{\mu}_q(\boldsymbol{x}) := W^q \boldsymbol{\phi}(\boldsymbol{x})$ and $\boldsymbol{\mu}_p(\boldsymbol{x}) := W^p \boldsymbol{\phi}(\boldsymbol{x})$. Then the log density ratio $r(\boldsymbol{y}|\boldsymbol{x}) = \log(q(\boldsymbol{y}|\boldsymbol{x})/p(\boldsymbol{y}|\boldsymbol{x}))$ is given by:

$$r(\boldsymbol{y}|\boldsymbol{x}) = \log\left( \sqrt{\frac{|\boldsymbol{\Lambda}_q|}{|\boldsymbol{\Lambda}_p|}} \frac{\exp(-(1/2)(\boldsymbol{y} - \boldsymbol{\mu}_q(\boldsymbol{x}))^{\mathrm{T}} \boldsymbol{\Lambda}_q(\boldsymbol{y} - \boldsymbol{\mu}_q(\boldsymbol{x})))}{\exp(-(1/2)(\boldsymbol{y} - \boldsymbol{\mu}_p(\boldsymbol{x}))^{\mathrm{T}} \boldsymbol{\Lambda}_p(\boldsymbol{y} - \boldsymbol{\mu}_p(\boldsymbol{x})))} \right) \tag{7}$$

$$= -\frac{1}{2}\boldsymbol{y}^{\mathrm{T}}\left(\boldsymbol{\Lambda}_q - \boldsymbol{\Lambda}_p\right)\boldsymbol{y} + \boldsymbol{y}^{\mathrm{T}}(\boldsymbol{\Lambda}_q W^q - \boldsymbol{\Lambda}_p W^p)\boldsymbol{\phi}(\boldsymbol{x}) + \psi(\boldsymbol{\phi}(\boldsymbol{x})), \tag{8}$$

where $\psi(\boldsymbol{\phi}(\boldsymbol{x}))$ represents the terms independent of $\boldsymbol{y}$. Now, if we assume that $\boldsymbol{\Lambda}_q = \boldsymbol{\Lambda}_p := \boldsymbol{\Lambda}$, we can ignore the quadratic term. If we further express $\boldsymbol{\Lambda}_q W^q - \boldsymbol{\Lambda}_p W^p$ in the form $V$, we can arrive at the form (3) again.

Indeed, however, the way that this regularization affects the training of the generator $G$ is a little unclear in its formulation. As we have repeatedly explained, our discriminator measures the divergence between the generator distribution $p$ and the target distribution $q$ on the assumption that $p(\boldsymbol{y}|\boldsymbol{x})$ and $q(\boldsymbol{y}|\boldsymbol{x})$ are relatively simple, and it is highly possible that we are gaining stability in the training process by imposing a regularity condition on the divergence measure. Meanwhile, however, the actual $p(\boldsymbol{y}|\boldsymbol{x})$ can only be implicitly derived from $p(\boldsymbol{x}, \boldsymbol{y})$ in computation, and can possibly take numerous forms other than the ones we have considered here. We must admit that there is a room here for an important *theoretical* work to be done in order to assess the relationship between the choice of the function space for the discriminator and training process of the generator.

## 4 COMPARISON WITH OTHER METHODS

As described above, (3) is a form that is true for frequently occurring situations. In contrast, incorporation of the conditional information by concatenation is rather arbitrary and can possibly include into the pool of candidate functions some sets of functions for which it is difficult to find a logical basis. Indeed, if the situation calls for multimodal $p(\boldsymbol{y}|\boldsymbol{x})$, it might be smart not to use the model that we suggest here. Otherwise, however, we expect our model to perform better; in general, it is preferable to use a discriminator that respects the presumed form of the probabilistic model.

Still another way to incorporate the conditional information into the training procedure is to directly manipulate the loss function. The algorithm of AC-GANs (Odena et al., 2017) use a discriminator $(D_1)$ that shares a part of its structure with the classifier($D_2$), and incorporates the label information into the objective function by augmenting the original discriminator objective with the likelihood score of the classifier on both the generated and training dataset (see Figure 1c). Plug and Play Generative models (PPGNs) (Nguyen et al., 2017) is another approach for the generative model that uses an auxiliary classifier function. It is a method that endeavors to make samples from $p(\boldsymbol{x}|y)$ using an MCMC sampler based on the Langevin equation with drift terms consisting of the gradient of an autoencoder prior $p(\boldsymbol{x})$ and a pretrained auxiliary classifier $p(y|\boldsymbol{x})$. With these method, one can generate a high quality image. However, these ways of using auxiliary classifier may unwittingly encourage the generator to produce images that are particularly easy for the auxiliary classifier to classify, and deviate the final $p(\boldsymbol{x}|y)$ from the true $q(\boldsymbol{x}|y)$. In fact, Odena et al. (2017) reports that this problem has a tendency to exacerbate with increasing number of labels. We were able to reproduce this phenomena in our experiments; when we implemented their algorithm on a dataset with 1000 class categories, the final trained model was able to generate only one image for most classes. Nguyen et al.'s PPGNs is also likely to suffer from the same problem because they are using an order of magnitude greater coefficient for the term corresponding to $p(y|\boldsymbol{x})$ than for the other terms in the Langevin equation.

## 5 EXPERIMENTS

In order to evaluate the effectiveness of our newly proposed architecture for the discriminator, we conducted two sets of experiments: class conditional image generation and super-resolution on ILSVRC2012 (ImageNet) dataset (Russakovsky et al., 2015). For both tasks, we used the ResNet (He et al., 2016b) based discriminator and the generator used in Gulrajani et al. (2017), and applied spectral normalization (Miyato et al., 2018) to the all of the weights of the discriminator to regularize the Lipschitz constant. For the objective function, we used the following *hinge* version

of the standard adversarial loss (1) (Lim & Ye, 2017; Tran et al., 2017)

$$L(\hat{G}, D) = E_{q(y)} \left[ E_{q(\boldsymbol{x}|y)} \left[ \max(0, 1 - D(\boldsymbol{x}, y)) \right] + E_{q(y)} \left[ E_{p(\boldsymbol{z})} \left[ \max\left(0, 1 + D\left(\hat{G}(\boldsymbol{z}, y), y\right)\right) \right] \right] \right],$$

$$L(G, \hat{D}) = -E_{q(y)} \left[ E_{p(\boldsymbol{z})} \left[ \hat{D}(G(\boldsymbol{z}, y), y) \right] \right], \tag{9}$$

where the last activation function $\mathcal{A}$ of $D$ is identity function. $p(\boldsymbol{z})$ is standard Gaussian distribution and $G(\boldsymbol{z}, y)$ is the generator network. For all experiments, we used Adam optimizer (Kingma & Ba, 2015) with hyper-parameters set to $\alpha = 0.0002, \beta_1 = 0, \beta_2 = 0.9$. We updated the discriminator five times per each update of the generator. We will use *concat* to designate the models (Figure 1b)[2], and use *projection* to designate the proposed model (Figure 1d) .

## 5.1 CLASS-CONDITIONAL IMAGE GENERATION

The ImageNet dataset used in the experiment of class conditional image generation consisted of 1,000 image classes of approximately 1,300 pictures each. We compressed each images to $128 \times 128$ pixels. Unlike for AC-GANs[3] we used a single pair of a ResNet-based generator and a discriminator. Also, we used conditional batch normalization (Dumoulin et al., 2017b; de Vries et al., 2017) for the generator. As for the architecture of the generator network used in the experiment, please see Figure 14 for more detail. Our proposed *projection* model discriminator is equipped with a 'projection layer' that takes inner product between the embedded one-hot vector $\boldsymbol{y}$ and the intermediate output (Figure 14a). As for the structure of the the *concat* model discriminator to be compared against, we used the identical bulk architecture as the *projection* model discriminator, except that we removed the projection layer from the structure and concatenated the spatially replicated embedded conditional vector $\boldsymbol{y}$ to the output of third ResBlock. We also experimented with AC-GANs as the current state of the art model. For AC-GANs, we placed the softmax layer classifier to the same structure shared by *concat* and *projection*. For each method, we updated the generator 450K times, and applied linear decay for the learning rate after 400K iterations so that the rate would be 0 at the end. For the comparative experiments, we trained the model for 450K iterations, which was ample for the training of *concat* to stabilize. AC-GANs collapsed prematurely before the completion of 450K iterations, so we reported the result from the peak of its performance ( 80K iterations). For all experiments throughout, we used the training over 450K iterations for comparing the performances. On a separate note, our method continued to improve even after 450K. We therefore also reported the inception score and FID of the extended training (850K iterations) for our method exclusively. See the table 1 for the exact figures.

We used inception score (Salimans et al., 2016) for the evaluation of the visual appearance of the generated images. It is in general difficult to evaluate how 'good' the generative model is. Indeed, however, either subjective or objective, some definite measures of 'goodness' exists, and essential two of them are 'diversity' and the sheer visual quality of the images. One possible candidate for quantitative measure of diversity and visual appearance is FID (Heusel et al., 2017). We computed FID between the generated images and dataset images within each class, and designated the values as intra FIDs. More precisely, FID (Heusel et al., 2017) measures the 2-Wasserstein distance between the two distributions $q_y$ and $p_y$, and is given by $F(q_y, p_y) = \|\boldsymbol{\mu}_{q_y} - \boldsymbol{\mu}_{p_y}\|_2^2 + \text{trace}\left(C_{q_y} + C_{p_y} - 2(C_{q_y} C_{p_y})^{1/2}\right)$, where $\{\boldsymbol{\mu}_{q_y}, C_{q_y}\}, \{\boldsymbol{\mu}_{p_y}, C_{p_y}\}$ are respectively the mean and the covariance of the final feature vectors produced by the inception model (Szegedy et al., 2015) from the true samples and generated samples of class $y$. When the set of generated examples have collapsed modes, the trace of $C_{p_y}$ becomes small and the trace term itself becomes large. In order to compute $C_{q_y}$ we used all samples in the training data belonging to the class of concern, and used 5000 generated samples for the computation of $C_{p_y}$. We empirically observed in our experiments that intra FID is, to a certain extent, serving its purpose well in measuring the diversity and the visual quality.

To highlight the effectiveness of our inner-product based approach (*projection*) of introducing the conditional information into the model, we compared our method against the state of the art AC-GANs as well as the conventional incorporation of the conditional information via concatenation

---

[2]in the preliminary experiments of the image geneation task on CIFAR-10 (Torralba et al., 2008) and CIFAR-100 (Torralba et al., 2008), we confirmed that hidden concatenation is better than input concatenation in terms of the inception scores. For more details, please see Table 3 in the appendix section.

[3]For AC-GANs, the authors prepared a pair of discriminator and generator for each set classes of size 10.

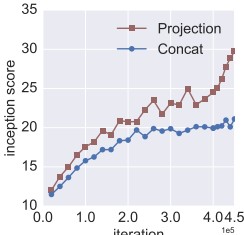

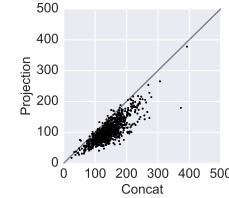

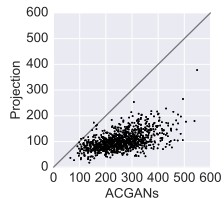

(a) *concat* vs *projection*     (b) AC-GANs vs *projection*

Figure 3: Learning curves of cGANs with *concat* and *projection* on ImageNet.

Figure 4: Comparison of intra FID scores for *projection* *concat*, and AC-GANs on ImageNet. Each dot corresponds to a class.

Table 1: Inception score and intra FIDs on ImageNet.

| Method | Inception Score | Intra FID |
|---|---|---|
| AC-GANs | 28.5±.20 | 260.0 |
| concat | 21.1±.35 | 141.2 |
| projection | **29.7**±.61 | **103.1** |
| *projection (850K iteration) | **36.8**±.44 | **92.4** |

at hidden layer (*concat*). As we can see in the training curves Figure 3, *projection* outperforms inception score than *concat* throughout the training. Table 1 compares the intra class FIDs and the inception Score of the images generated by each method. The result shown here for the AC-GANs is that of the model at its prime in terms of the inception score, because the training collapsed at the end. We see that the images generated by *projection* have lower intra FID scores than both adversaries, indicating that the Wasserstein distance between the generative distribution by *projection* to the target distribution is smaller. For the record, our model performed better than other models on the CIFAR10 and CIFAR 100 as well (See Appendix A).

Figure 10a and 10b shows the set of classes for which (a) *projection* yielded results with better intra FIDs than the *concat* and (b) the reverse. From the top, the figures are listed in descending order of the ratio between the intra FID score between the two methods. Note that when the *concat* outperforms *projection* it only wins by a slight margin, whereas the *projection* outperforms *concat* by large margin in the opposite case. A quick glance on the cases in which the *concat* outperforms the *projection* suggests that the FID is in fact measuring the visual quality, because both sets looks similar to the human eyes in terms of appearance. Figure 5 shows an arbitrarily selected set of results yielded by AC-GANs from variety of $z$s. We can clearly observe the mode-collapse on this batch. This is indeed a tendency reported by the inventors themselves (Odena et al., 2017). AC-GANs can generate easily recognizable (i.e classifiable) images, but at the cost of losing diversity and hence at the cost of constructing a generative distribution that is significantly different from the target distribution as a whole. We can also assess the low FID score of *projection* from different perspective. By construction, the trace term of intra FID measures the degree of diversity within the class. Thus, our result on the intra FID scores also indicates that that our *projection* is doing better in reproducing the diversity of the original. The GANs with the *concat* discriminator also suffered from mode-collapse for some classes (see Figure 6). For the set of images generated by *projection*, we were not able to detect any notable mode-collapse.

Figure 7a shows the samples generated with the *projection* model for the classes on which the cGAN achieved lowest intra FID scores (that is the classes on which the generative distribution were particularly close to target conditional distribution), and Figure 7b the reverse. While most of the images listed in Figure 7a are of relatively high quality, we still observe some degree of mode-collapse. Note that the images in the classes with high FID are featuring complex objects like human; that is, one can expect the diversity within the class to be wide. However, we note that we did not use the most complicated neural network available for the experiments presented on this paper, because we prioritized the completion of the training within a reasonable time frame. It is very possible that, by increasing the complexity of the model, we will be able to further improve the

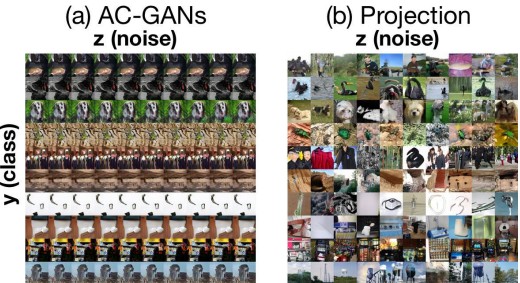

Figure 5: comparison of the images generated by (a) AC-GANs and (b) *projection*.

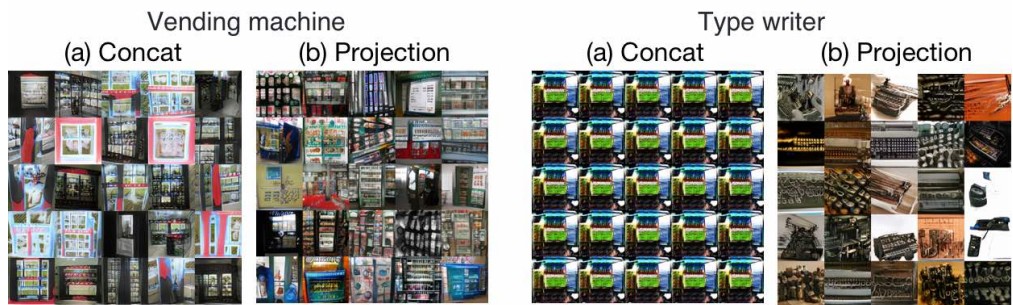

Figure 6: Collapsed images on the *concat* model.

visual quality of the images and the diversity of the distribution. In Appendix D, we list images of numerous classes generated by cGANs trained with our *projection* model.

**Category Morphing**   With our new architecture, we were also able to successfully perform category morphism. When there are classes $y_1$ and $y_2$, we can create an interpolated generator by simply mixing the parameters of conditional batch normalization layers of the conditional generator corresponding to these two classes. Figure 8 shows the output of the interpolated generator with the same $z$. Interestingly, the combination is also yielding meaningful images when $y_1$ and $y_2$ are significantly different.

**Fine-tuning with the pretrained model on the ILSVRC2012 classification task.**   As we mentioned in Section 4, the authors of Plug and Play Generative model (PPGNs) (Nguyen et al., 2017) were able to improve the visual appearance of the model by augmenting the cost function with that of the label classifier. We also followed their footstep and augmented the original generator loss with an additional auxiliary classifier loss. As warned earlier regarding this type of approach, however, this type of modification tends to only improve the visual performance of the images that are easy for the pretrained model to classify. In fact, as we can see in Appendix B, we were able to improve the visual appearance the images with the augmentation, but at the cost of diversity.

## 5.2   SUPER-RESOLUTION

We also evaluated the effectiveness of (3) in its application to the super-resolution task. Put formally, the super-resolution task is to infer the high resolution RGB image of dimension $\boldsymbol{x} \in \mathbb{R}^{R_H \times R_H \times 3}$ from the low resolution RGB image of dimension $\boldsymbol{y} \in \mathbb{R}^{R_L \times R_L \times 3}$; $R_H > R_L$. This task is very much the case that we presumed in our model construction, because $p(\boldsymbol{y}|\boldsymbol{x})$ is most likely unimodal even if $p(\boldsymbol{x}|\boldsymbol{y})$ is multimodal. For the super-resolution task, we used the following formulation for discriminator function:

$$f(\boldsymbol{x}, \boldsymbol{y}; \theta) = \sum_{i,j,k} \left( y_{ijk} F_{ijk}(\boldsymbol{\phi}(\boldsymbol{x}; \theta_\Phi)) \right) + \psi(\boldsymbol{\phi}(\boldsymbol{x}; \theta_\Phi); \theta_\Psi), \tag{10}$$

where $\boldsymbol{F}(\boldsymbol{\phi}(\boldsymbol{x}; \theta_\Phi)) = V * \boldsymbol{\phi}(\boldsymbol{x}; \theta_\Phi)$ where $V$ is a convolutional kernel and $*$ stands for convolution operator. Please see Figure 15 in the appendix section for the actual network architectures we used

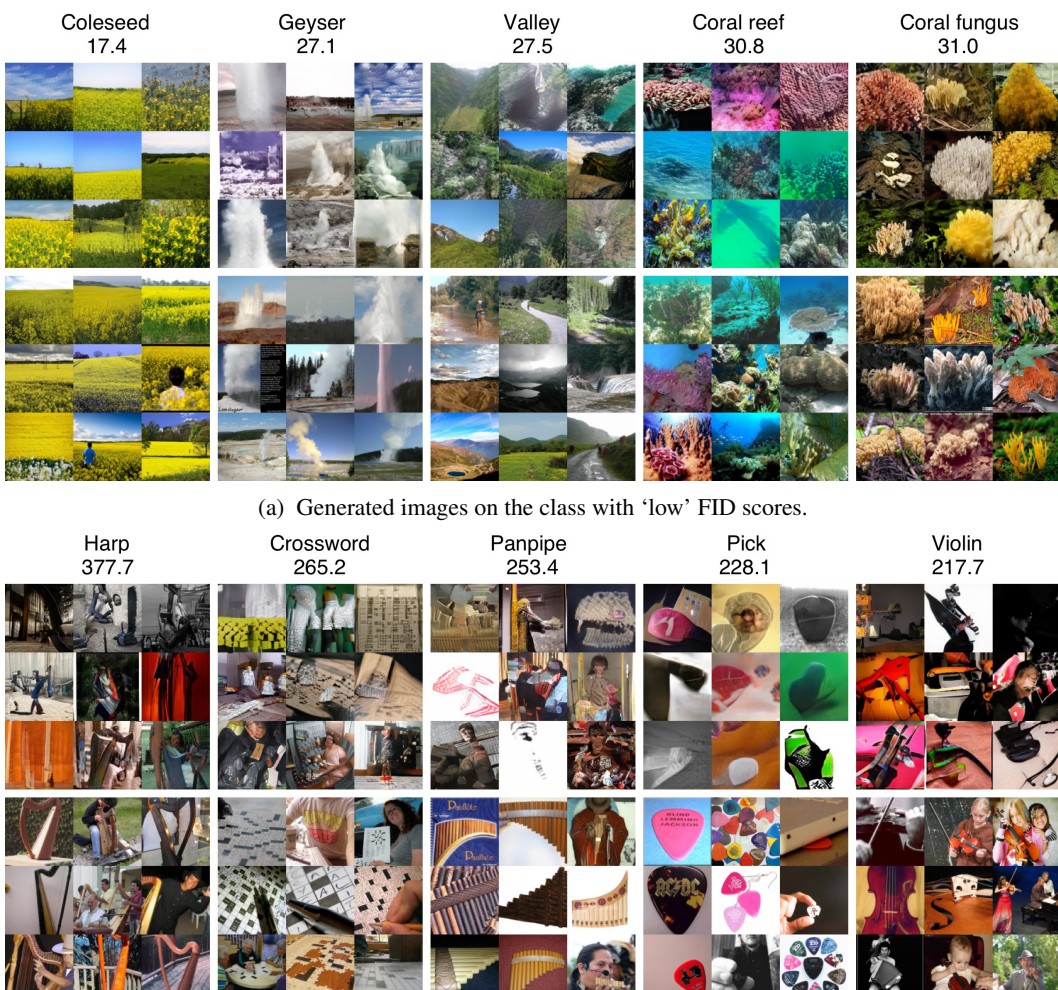

| Coleseed | Geyser | Valley | Coral reef | Coral fungus |
|----------|--------|--------|------------|--------------|
| 17.4 | 27.1 | 27.5 | 30.8 | 31.0 |

(a) Generated images on the class with 'low' FID scores.

| Harp | Crossword | Panpipe | Pick | Violin |
|------|-----------|---------|------|--------|
| 377.7 | 265.2 | 253.4 | 228.1 | 217.7 |

(b) generated images on the class with 'high' FID scores.

Figure 7: 128×128 pixel images generated by the *projection* method for the classes with (a) bottom five FID scores and (b) top five FID scores. The string and the value above each panel are respectively the name of the corresponding class and the FID score. The second row in each panel corresponds to the original dataset.

for this task. For this set of experiments, we constructed the *concat* model by removing the module in the *projection* model containing the the inner product layer and the accompanying convolution layer altogether, and simply concatenated $y$ to the output of the ResBlock preceding the inner product module in the original. As for the resolutions of the image datasets, we chose $R_H = 128$ and $R_L = 32$, and created the low resolution images by applying bilinear downsampling on high resolution images. We updated the generators 150K times for all methods, and applied linear decay for the learning rate after 100K iterations so that the final learning rate was 0 at 150K-th iteration.

Figure 9 shows the result of our super-resolution. The bicubic super-resolution is very blurry, and *concat* result is suffering from excessively sharp and rough edges. On the other hand, the edges of the images generated by our *projection* method are much clearer and smoother, and the image itself is much more faithful to the original high resolution images. In order to qualitatively compare the performances of the models, we checked MS-SSIM (Wang et al., 2003) and the classification accuracy of the inception model on the generated images using the validation set of the ILSVRC2012 dataset. As we can see in Table 2, our *projection* model was able to achieve high inception accuracy and high MS-SSIM when compared to *bicubic* and *concat*. Note that the performance of super-resolution with *concat* model even falls behind those of the bilinear and bicubic super-resolutions

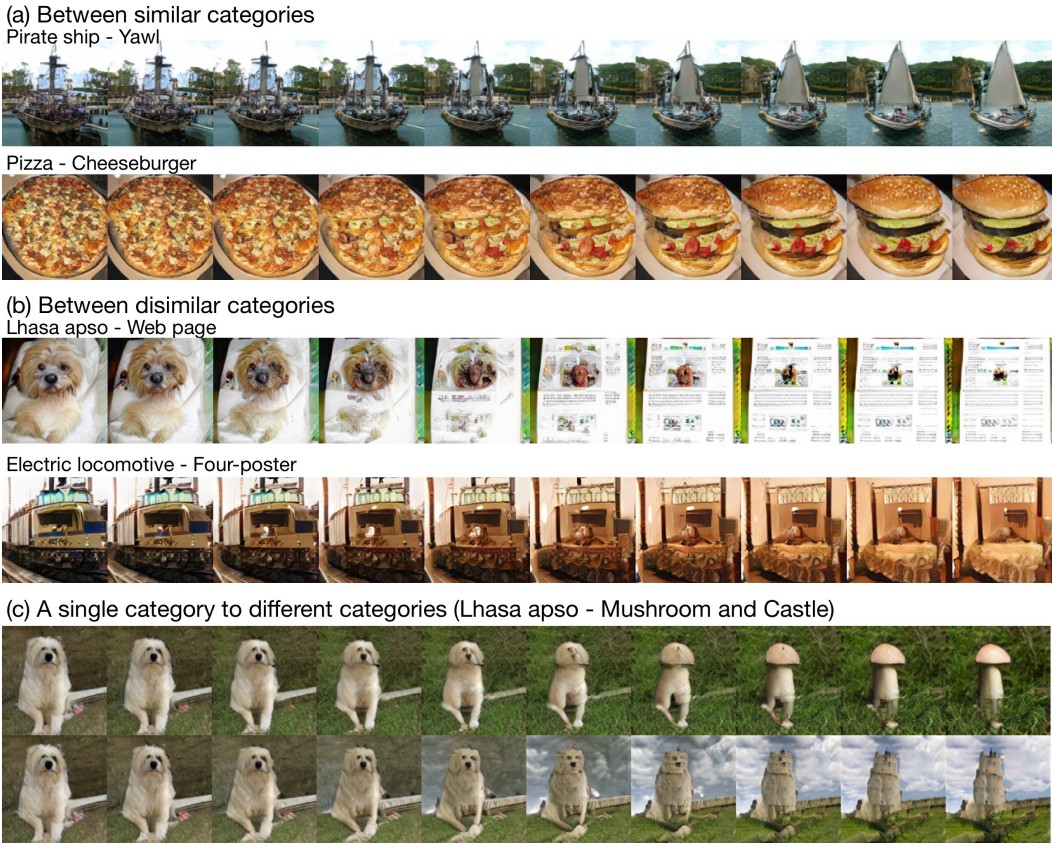

Figure 8: Category morphing. More results are in the appendix section.

Table 2: Inception accuracy and MS-SSIM on different super-resolution methods. We picked up dataset images from the validation set.

| Method | biliear | bicubic | concat | projection | projection (10 MC) |
|---|---|---|---|---|---|
| Inception Acc.(%) | 23.1 | 31.4 | 11.0 | **35.2** | **36.4** |
| MS-SSIM | 0.835 | 0.859 | 0.829 | **0.878** | - |

in terms of the inception accuracy. Also, we used *projection* model to generate multiple batches of images with different random values of $z$ to be fed to the generator and computed the average of the logits of the inception model on these batches (MC samples). We then used the so-computed average logits to make prediction of the labels. With an ensemble over 10 seeds (10 MC in Table 2), we were able to improve the inception accuracy even further. This result indicates that our GANs are learning the super-resolution as an distribution, as opposed to deterministic function. Also, the success with the ensemble also suggests a room for a new way to improve the accuracy of classification task on low resolution images.

## 6 CONCLUSION

Any specification on the form of the discriminator imposes a regularity condition for the choice for the generator distribution and the target distribution. In this research, we proposed a model for the discriminator of cGANs that is motivated by a commonly occurring family of probabilistic models. This simple modification was able to significantly improve the performance of the trained generator on conditional image generation task and super-resolution task. The result presented in this paper is strongly suggestive of the importance of the choice of the form of the discriminator and the design

| Low resolution | (a) Bicubic | (b) GANs, concat | (c) GANs, projection | High resolution (Ground truth) |

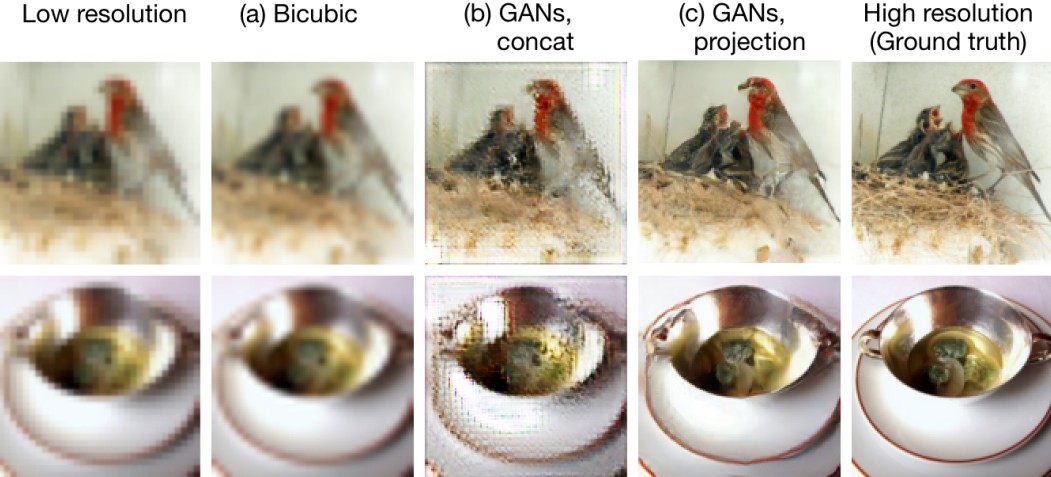

Figure 9: 32x32 to 128x128 super-resolution by different methods

of the distributional metric. We plan to extend this approach to other applications of cGANs, such as semantic segmentation tasks and image to image translation tasks.

### ACKNOWLEDGMENTS

We would like to thank the members of Preferred Networks, Inc., especially Richard Calland, Sosuke Kobayashi and Crissman Loomis, for helpful comments. We would also like to thank Shoichiro Yamaguchi, a graduate student of Kyoto University, for helpful comments.

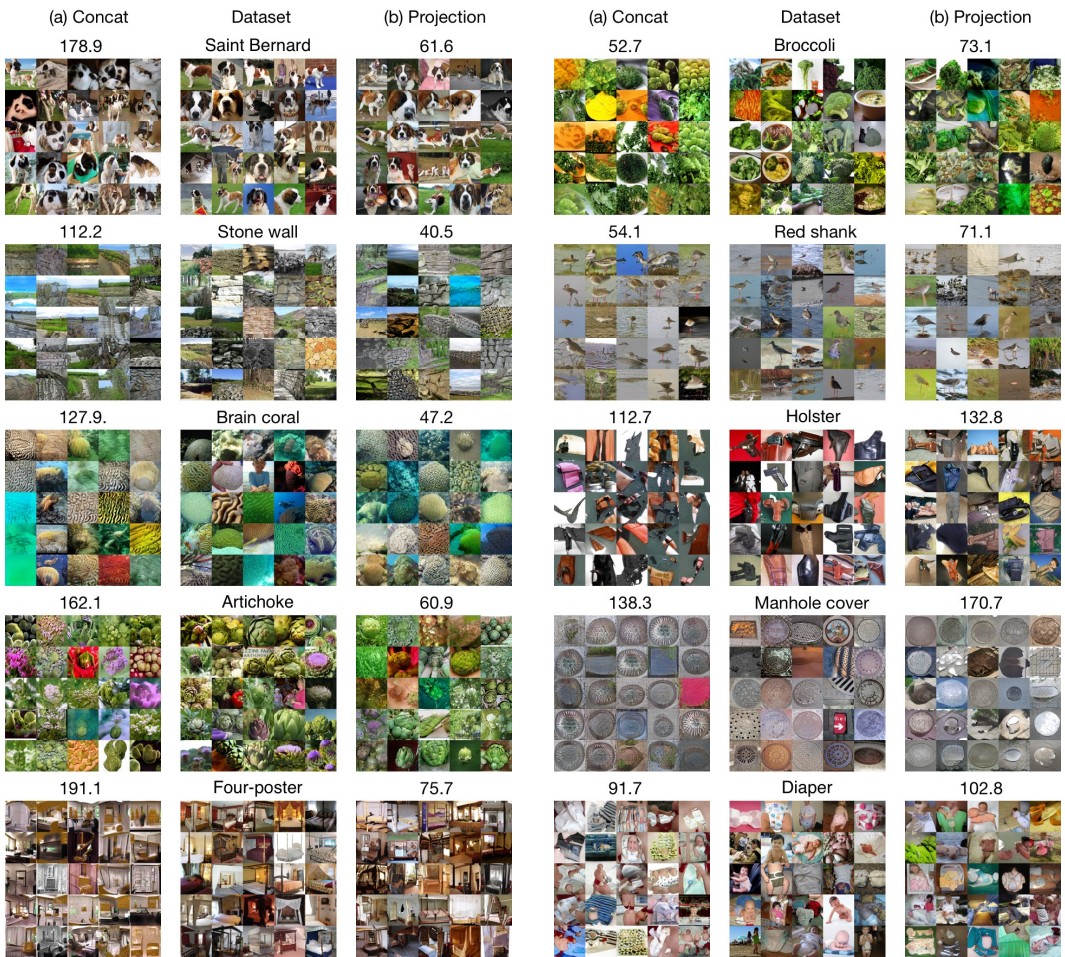

(a) Images better with *projection* than *concat*.  (b) Images better with *concat* than *projection*.

Figure 10: Comparison of *concat* vs. *projection*. The value attached above each panel represents the achieved FID score.

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

## A RESULTS OF CLASS CONDITIONAL IMAGE GENERATION ON CIFAR-10 AND CIFAR-100

As a preliminary experiment, we compared the performance of conditional image generation on CIFAR-10 and CIFAR-100 3. For the discriminator and the generator, we reused the same architecture used in Miyato et al. (2018) for the task on CIFAR-10. For the adversarial objective functions, we used (9), and trained both machine learners with the same optimizer with same hyper parameters we used in Section 5. For our *projection* model, we added the projection layer to the discriminator in the same way we did in the ImageNet experiment (before the last linear layer). Our *projection* model achieved better performance than other methods on both CIFAR-10 and CIFAR-100. Concatenation at hidden layer (*hidden concat*) was performed on the output of second ResBlock of the discriminator. We tested *hidden concat* as a comparative method in our main experiments on ImageNet, because the concatenation at hidden layer performed better than the concatenation at the input layer (*input concat*) when the number of classes was large (CIFAR-100).

To explore how the hyper-parameters affect the performance of our proposed architecture, we conducted hyper-parameter search on CIFAR-100 about the Adam hyper-parameters (learning rate $\alpha$ and 1st order momentum $\beta_1$) for both our proposed architecture and the baselines. Namely, we varied each one of these parameters while keeping the other constant, and reported the inception scores for all methods including several versions of *concat* architectures to compare. We tested with *concat* module introduced at (a) input layer, (b) hidden layer, and at (c) output layer. As we can see in Figure 11, our *projection* architecture excelled over all other architectures for all choice of the parameters, and achieved the inception score of 9.53. Meanwhile, *concat* architectures were able to achieve all 8.82 at most. The best *concat* model in term of the inception score on CIFAR-100 was the hidden *concat* with $\alpha = 0.0002$ and $\beta_1 = 0$, which turns out to be the very choice of the parameters we picked for our ImageNet experiment.

Table 3: The performance of class conditional image generation on CIFAR-10 (C10) and CIFAR-100 (C100).

| Method | Inception score | | FID | |
|---|---|---|---|---|
| | C10 | C100 | C10 | C100 |
| (Real data) | 11.24 | 14.79 | 7.60 | 8.94 |
| AC-GAN | 8.22 | 8.80 | 19.7 | 25.4 |
| input concat | 8.25 | 7.93 | 19.2 | 31.4 |
| hidden concat | 8.14 | 8.82 | 19.2 | 24.8 |
| (ours) projection | **8.62** | **9.04** | **17.5** | **23.2** |

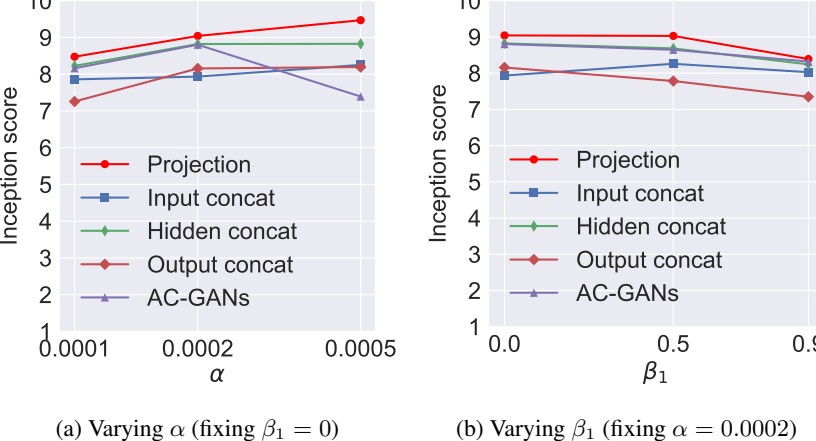

(a) Varying $\alpha$ (fixing $\beta_1 = 0$)     (b) Varying $\beta_1$ (fixing $\alpha = 0.0002$)

Figure 11: Inception scores on CIFAR-100 with different discriminator models varying hyper-parameters ($\alpha$ and $\beta_1$) of Adam optimizer.

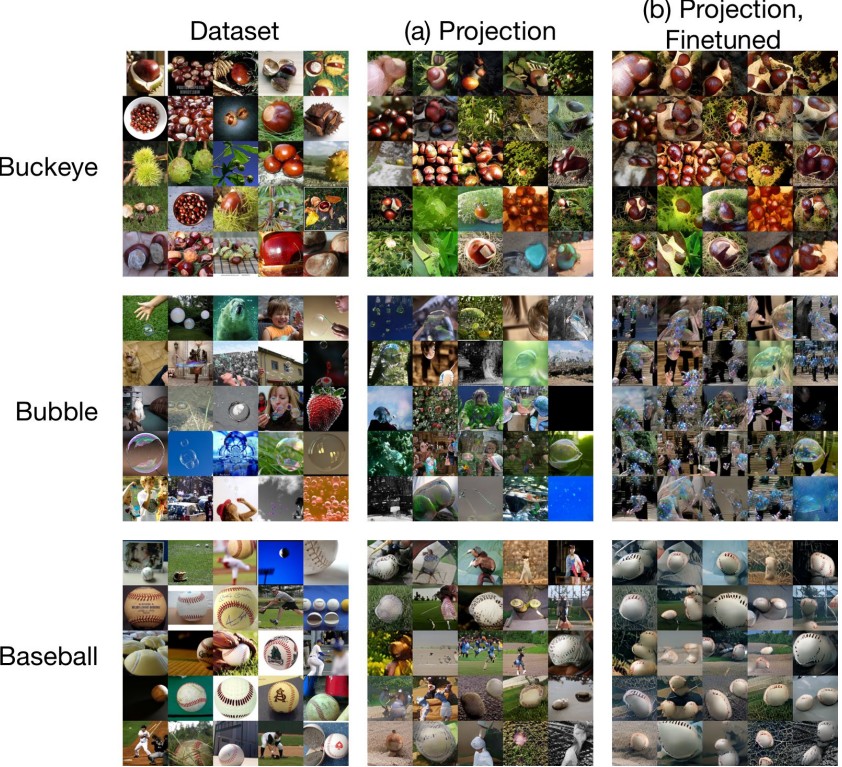

Figure 12: Effect of the finetuning with auxiliary classifier loss. Same coordinate in panel (a) and (b) corresponds to same value of $z$

Table 4: Inception score and intra FIDs on ImageNet with a pretrained model on classification tasks for ILSVRC2012 dataset. ‡Nguyen et al. (2017)

| Method | Inception Score | Intra FID |
|---|---|---|
| PPGNs‡ | 47.4 | N/A |
| projection(finetuned) | 210 | 54.2 |

## B    OBJECTIVE FUNCTION WITH AN AUXILIARY CLASSIFIER COST

In this experiment, we followed the footsteps of Plug and Play Generative model (PPGNs) (Nguyen et al., 2017) and augmented the original generator loss with an additional auxiliary classifier loss. In particular, we used the losses given by :

$$L\left(G, \hat{D}, \hat{p}_{\text{pre}}(y|\boldsymbol{x})\right) = -E_{q(y)}\left[E_{p(\boldsymbol{z})}\left[\hat{D}(G(\boldsymbol{z}, y), y) - L_C(\hat{p}_{\text{pre}}(y|G(\boldsymbol{z}, y)))\right]\right], \qquad (11)$$

where $\hat{p}_{\text{pre}}(y|\boldsymbol{x})$ is the fixed model pretrained for ILSVRC2012 classification task. For the actual experiment, we trained the generator with the original adversarial loss for the first 400K updates, and used the augmented loss for the last 50K updates. For the learning rate hyper parameter, we adopted the same values as other experiments we described above. For the pretrained classifier, we used ResNet50 model used in He et al. (2016a). Figure 12 compares the results generated by vanilla objective function and the results generated by the augmented objective function. As we can see in Table 4, we were able to significantly outperform PPGNs in terms of inception score. However, note that the images generated here are images that are easy to classify. The method with auxiliary classifier loss seems effective in improving the visual appearance, but not in training faithful generative model.

## C  MODEL ARCHITECTURES

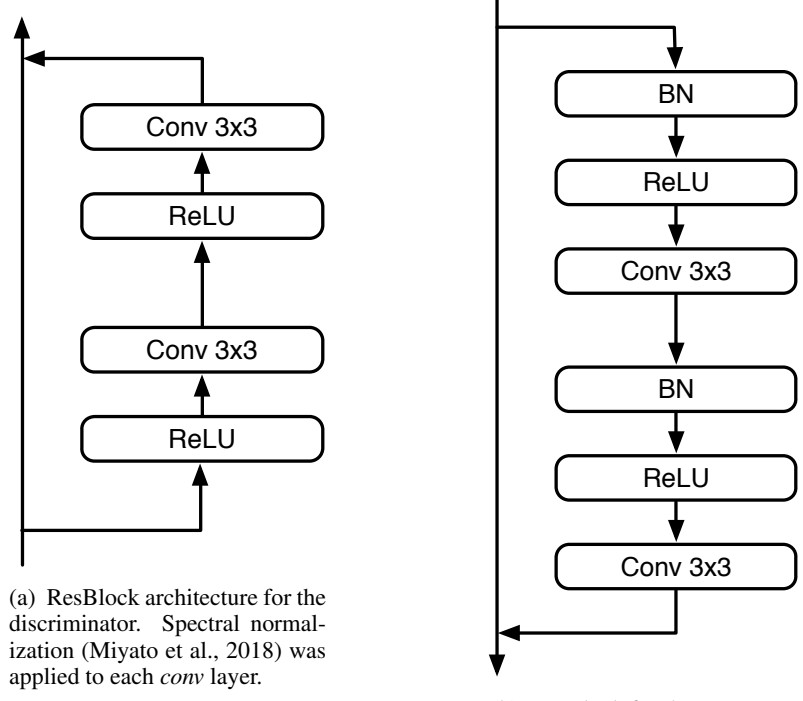

(a) ResBlock architecture for the discriminator. Spectral normalization (Miyato et al., 2018) was applied to each *conv* layer.

(b) ResBlock for the generator.

Figure 13: Architecture of the ResBlocks used in all experiments. For the generator generator's Resblock, conditional batch normalization layer (Dumoulin et al., 2017b; de Vries et al., 2017) was used in place of the standard batch normalization layer. For the ResBlock in the generator for the super resolution tasks that implements the upsampling, the random vector $z$ was fed to the model by concatenating the vector to the embedded low resolution image vector $y$ prior to the first convolution layer within the block. For the procedure of downsampling and upsampling, we followed the implementation by Gulrajani et al. (2017). For the discriminator, we performed downsampling (average pool) after the second *conv* of the ResBlock. For the generator, we performed upsampling before the first *conv* of the ResBlock. For the ResBlock that is performing the downsampling, we replaced the identity mapping with 1x1 *conv* layer followed by downsampling to balance the dimension. We did the essentially same for the Resblock that is performing the upsampling, except that we applied the upsampling before the 1x1 *conv*.

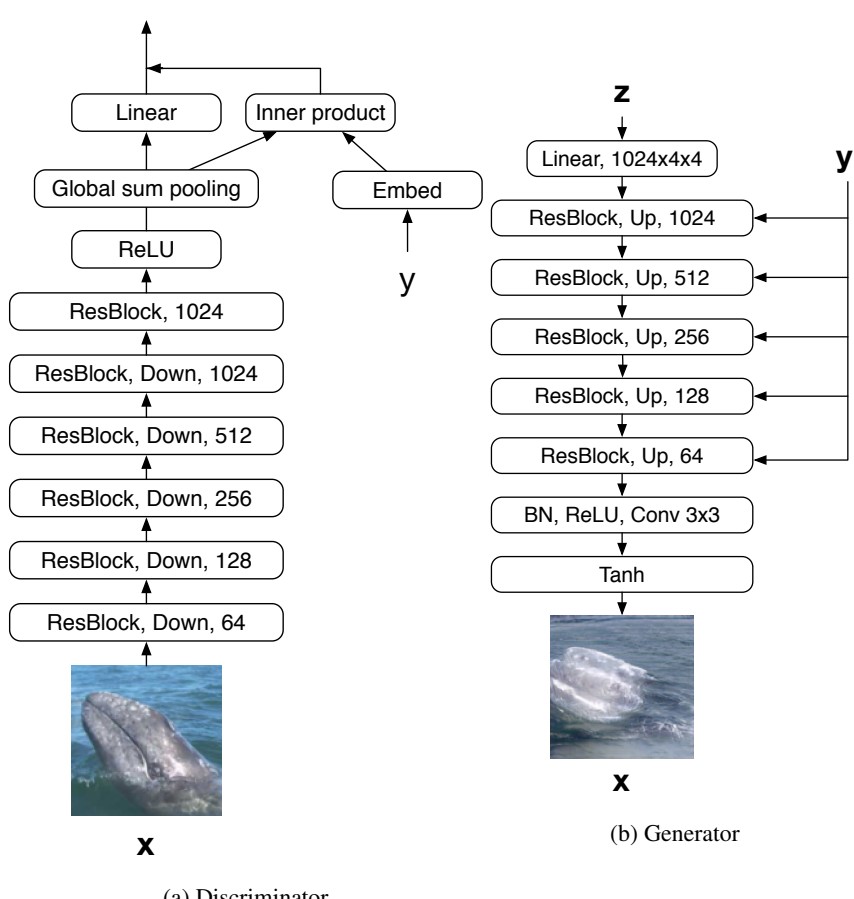

(a) Discriminator

(b) Generator

Figure 14: The models we used for the conditional image generation task.

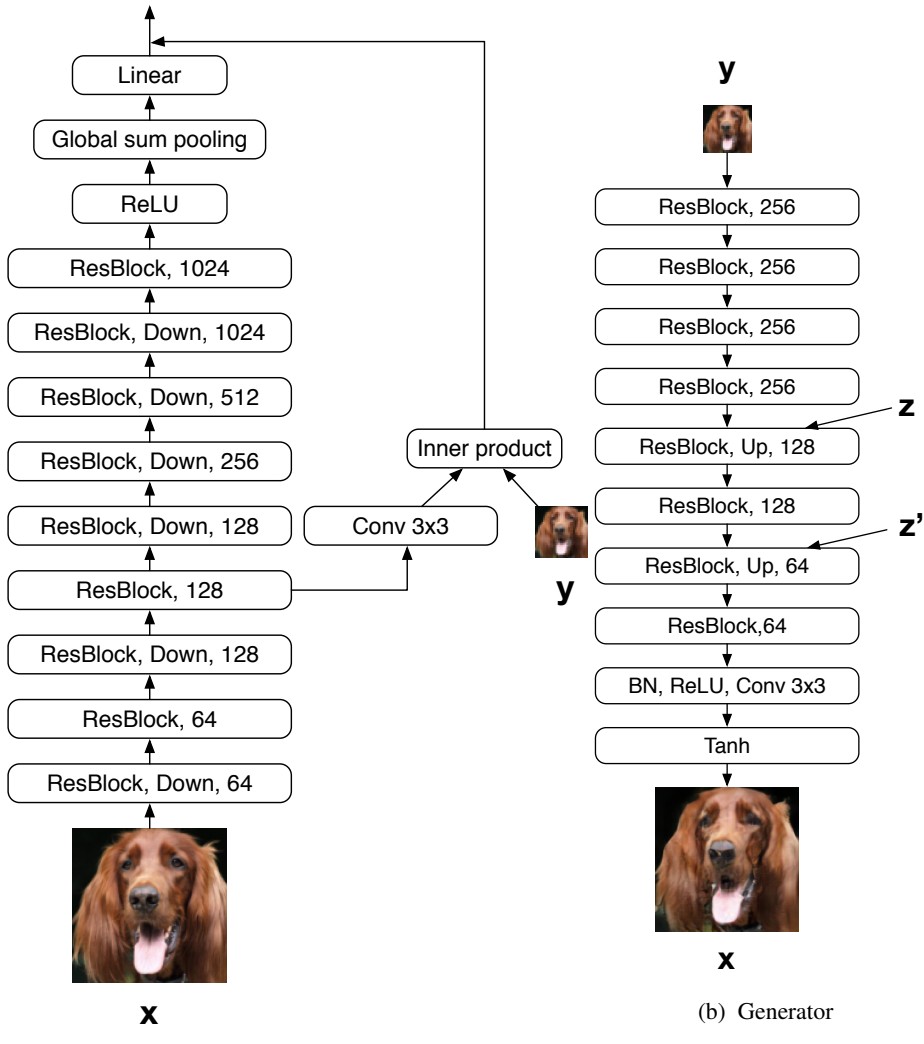

(a) Discriminator

(b) Generator

Figure 15: The models we used for the super resolution task.

# D    MORE RESULTS ON THE CONDITIONAL IMAGE GENERATION TASK

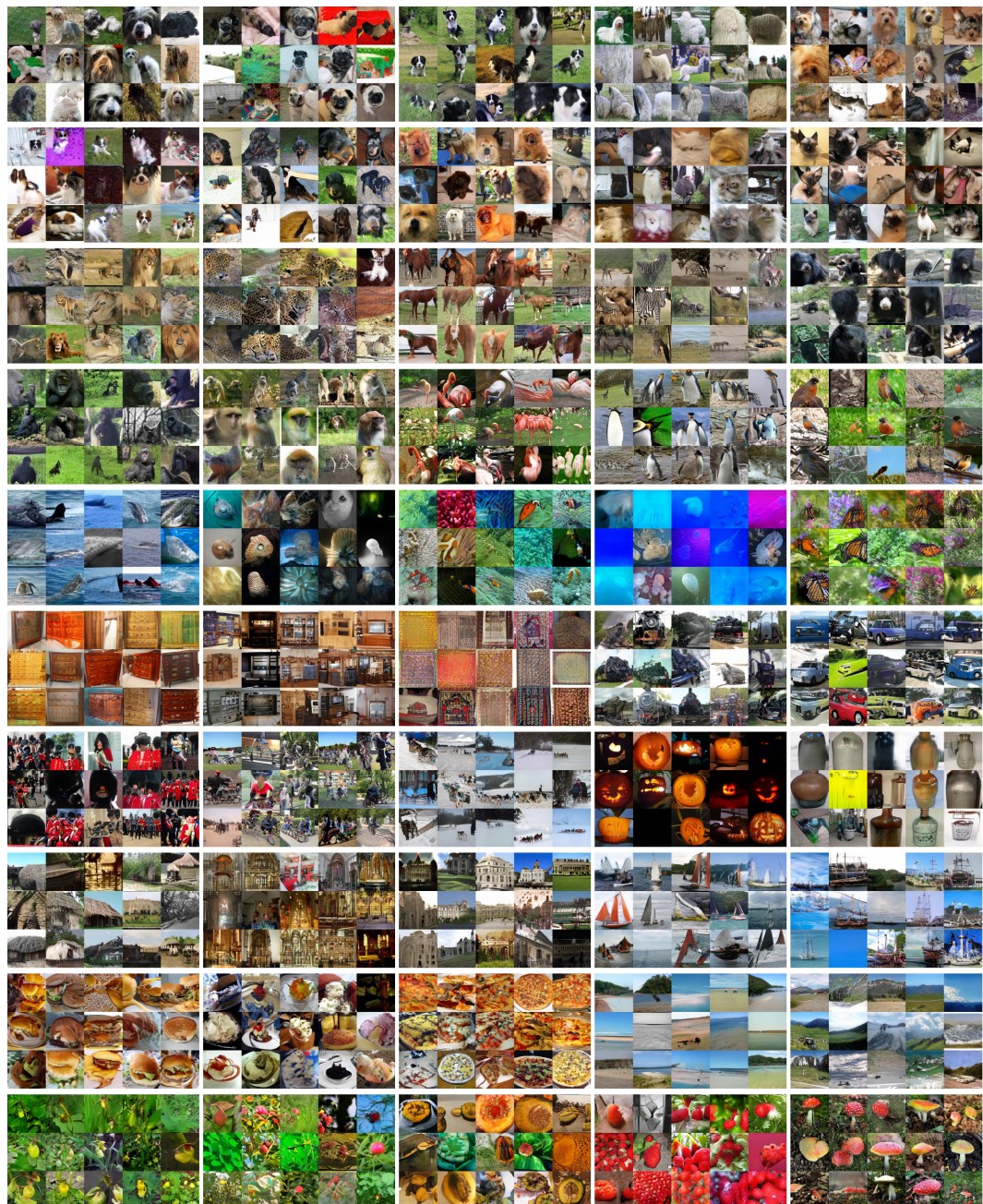

Figure 16:  128×128 generated examples on various categories with *projection* architecture. Each panel corresponds to a class.

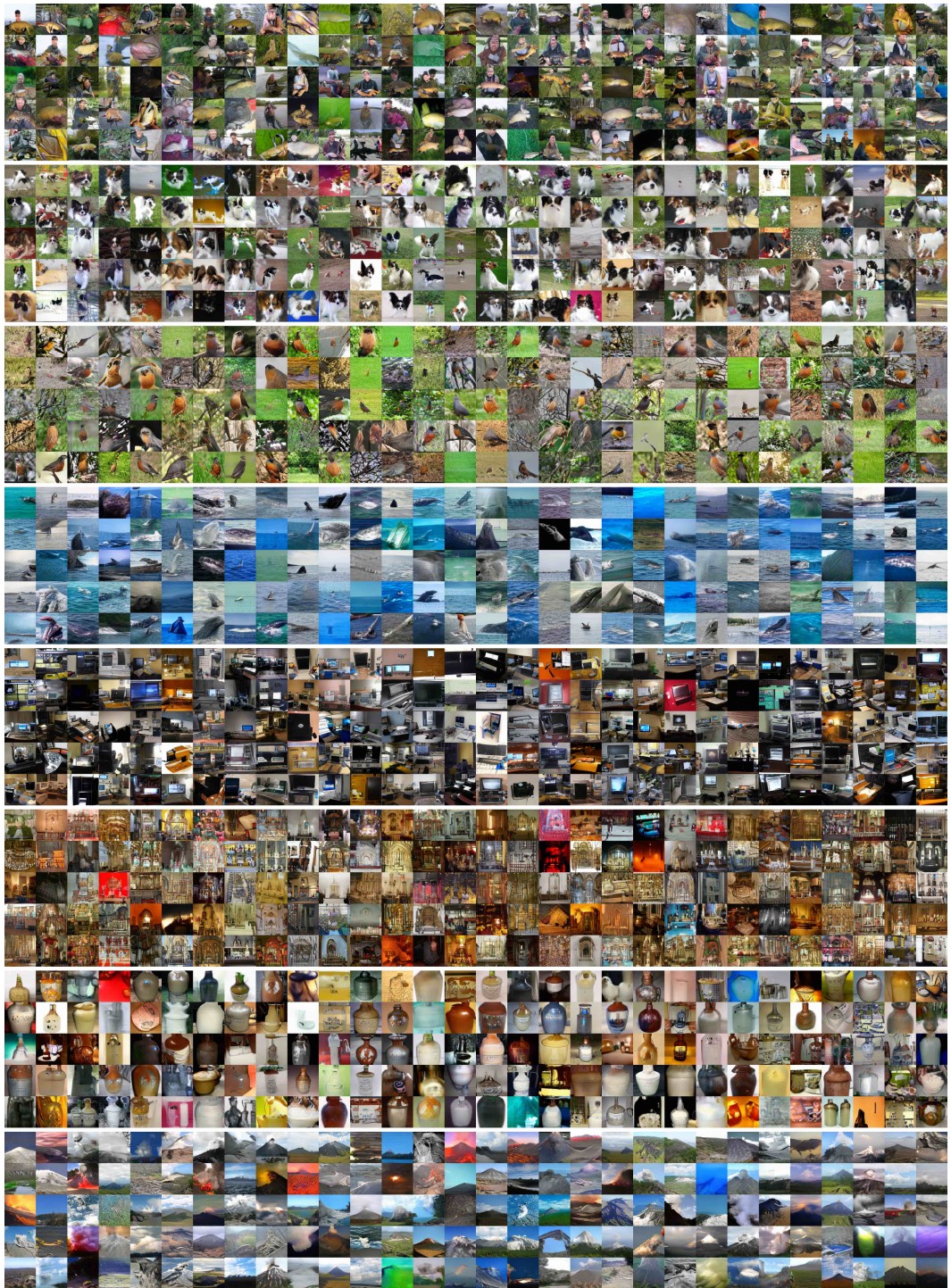

Figure 17: 128×128 generated examples on various categories with *projection* architecture. Each panel corresponds to a class. From top to bottom, tench, papillon, grey whale, desktop computer, altar, whiskey jug and volcano.

# E    RESULTS OF CATEGORY MORPHING

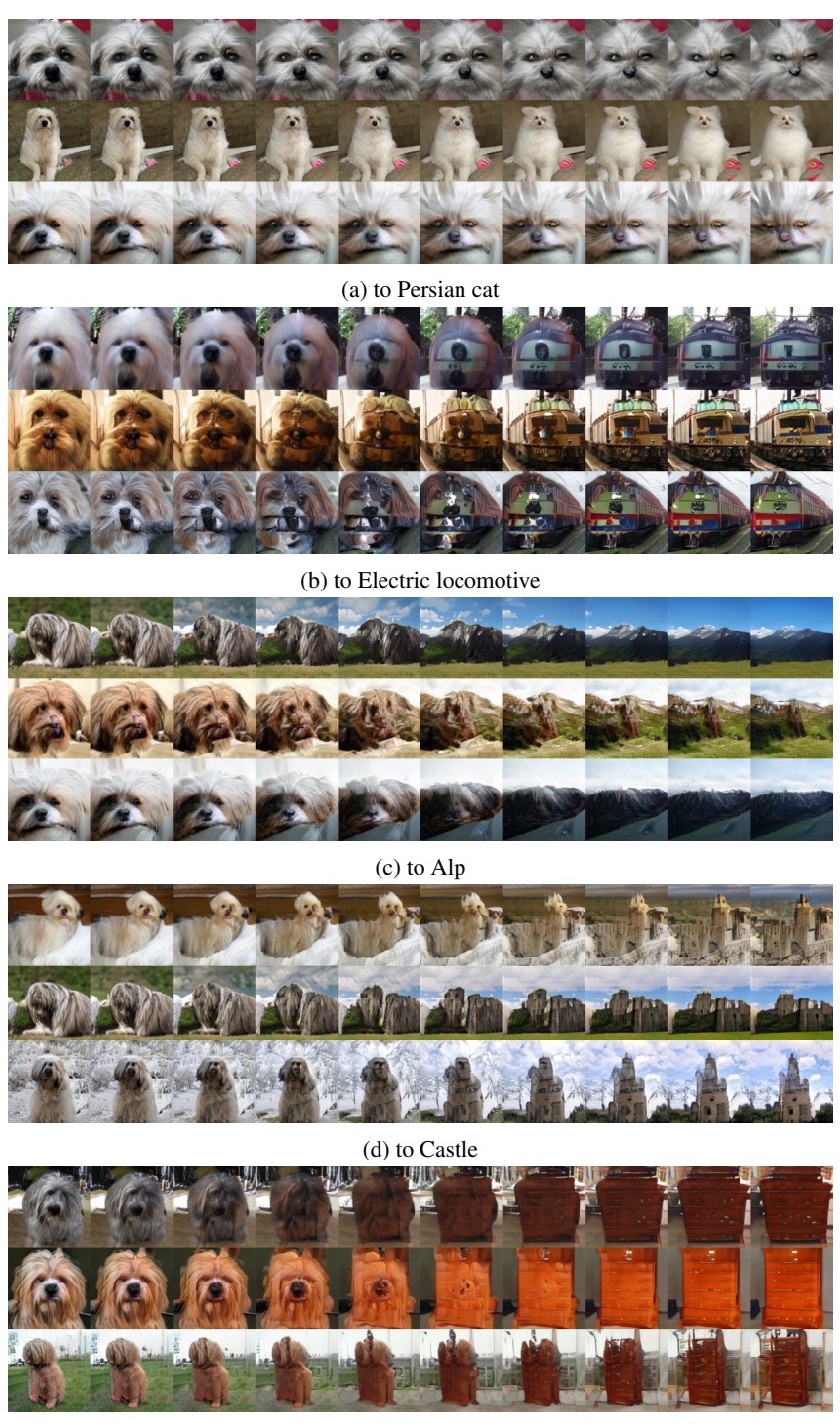

(a) to Persian cat

(b) to Electric locomotive

(c) to Alp

(d) to Castle

(e) to Chiffonier

Figure 18: Dog (Lhasa apso) to different categories

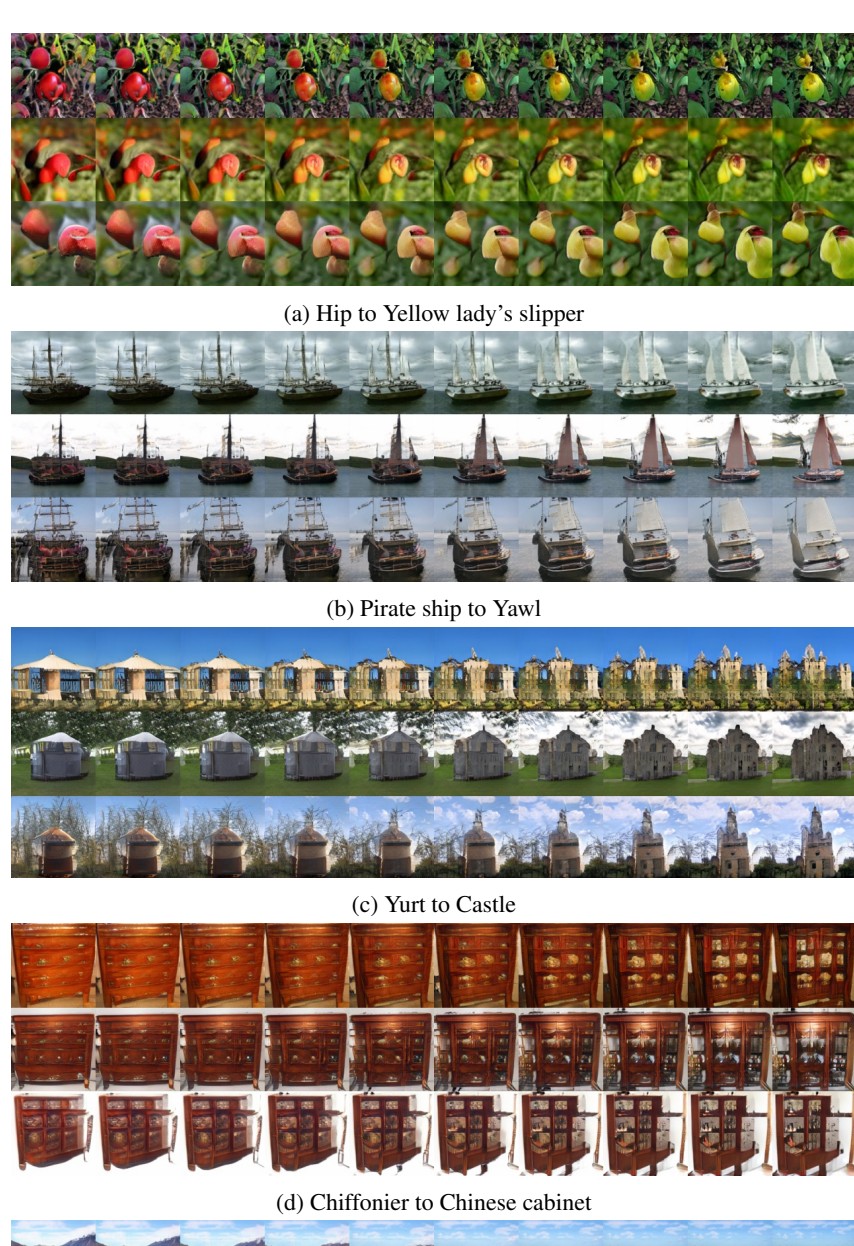

(a) Hip to Yellow lady's slipper

(b) Pirate ship to Yawl

(c) Yurt to Castle

(d) Chiffonier to Chinese cabinet

(e) Valley to Sandbar

Figure 19: Morphing between different categories

## F MORE RESULTS WITH SUPER-RESOLUTION

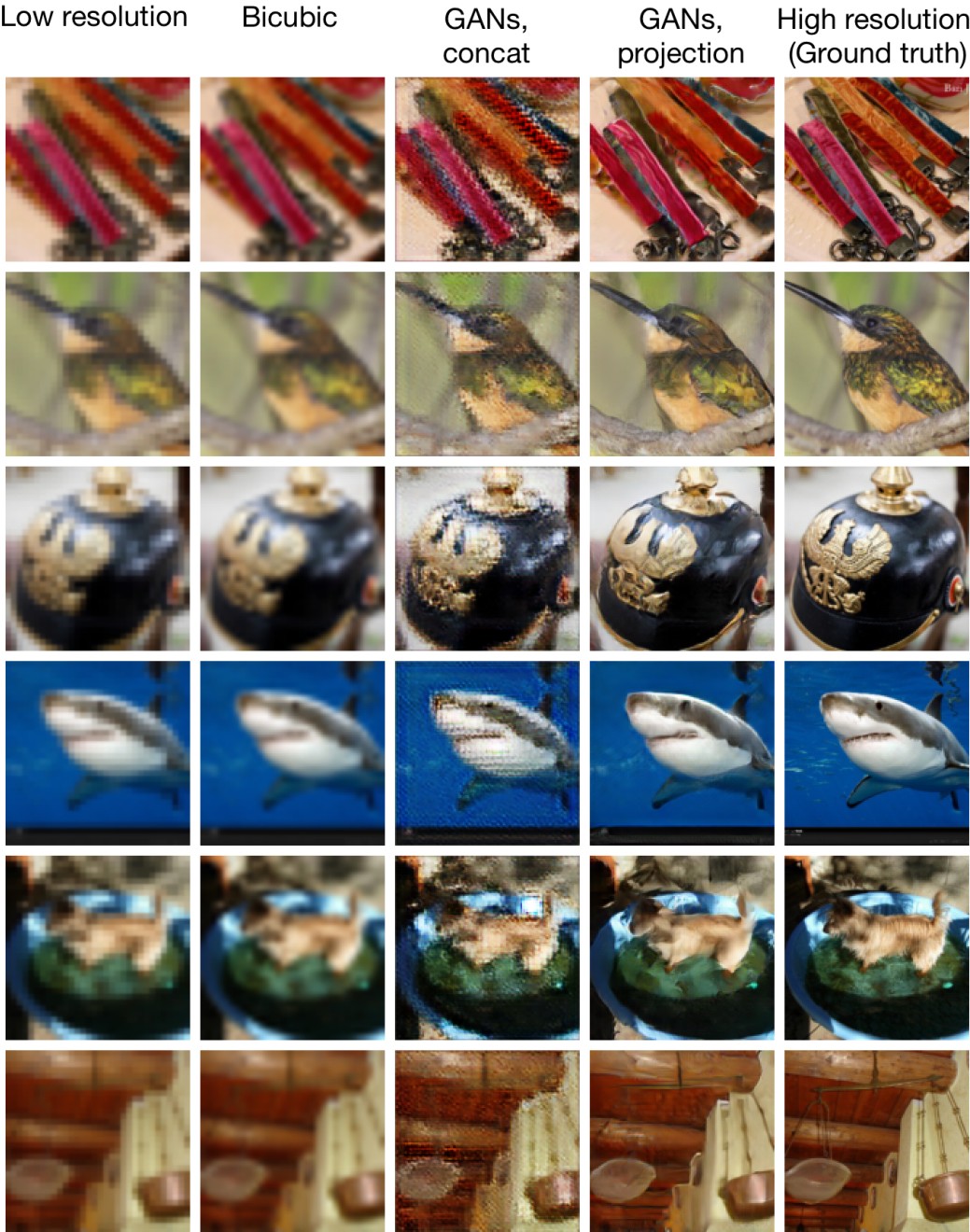

Figure 20: 32x32 to 128x128 super-resolution results

