# OpenReview forum: "cGANs with Projection Discriminator"
_ICLR.cc/2018/Conference — Accept (Poster)_

### Official Review · AnonReviewer3 · 2017-11-18
**Review of cGAN with projection discriminator**

**Rating:** 6
**Confidence:** 4

**Review:**


I thank the authors for the thoughtful response and updated manuscript. After reading through both, my review score remains unchanged.

=================

The authors describe a new variant of a generative adversarial network (GAN) for generating images. This model employs a 'projection discriminator' in order to incorporate image labels and demonstrate that the resulting model outperforms state-of-the-art GAN models.

Major comments:
1) Spatial resolution. What spatial resolution is the model generating images at? The AC-GAN work performed an analysis to assess how information is being introduced at each spatial resolution by assessing the gains in the Inception score versus naively resizing the image. It is not clear how much the gains of this model is due to generating better lower resolution images and performing simple upscaling. It would be great to see the authors address this issue in a serious manner.

2) FID in real data. The numbers in Table 1 appear favorable to the projection model. Please add error bars (based on Figure 4, I would imagine they are quite large). Additionally, would it be possible to compute this statistic for *real* images? I would be curious to know what the FID looks like as a 'gold standard'.

3) Conditional batch normalization.  I am not clear how much of the gains arose from employing conditional batch normalization versus the proposed method for incorporating the projection based discriminator. The former has been seen to be quite powerful in accomodating multi-modal tasks (e.g. https://arxiv.org/abs/1709.07871, https://arxiv.org/abs/1610.07629
). If the authors could provide some evidence highlighting the marginal gains of one technique, that would be extremely helpful.

Minor comments:
- I believe you have the incorrect reference for conditional batch normalization on Page 5.
A Learned Representation For Artistic Style
Dumoulin, Shlens and Kudlur (2017)
https://arxiv.org/abs/1610.07629

- Please enlarge images in Figure 5-8. Hard to see the detail of 128x128 images.

- Please add citations for Figures 1a-1b. Do these correspond with some known models?

Depending on how the authors respond to the reviews, I would consider upgrading the score of my review.

---

> ### Author Response · Authors · 2017-12-03
> **Response to AnonReviewer3**
>
>
> > 1) Spatial resolution. What spatial resolution is the model generating images at?
>
> We are sorry for the lack of the spatial resolution information. The model generates at "128x128" spatial resolution.
>
>
> > 1) Spatial resolution,  3) Conditional batch normalization,
>
> The goal of our paper is to show the efficacy of our "projection" model for the discriminator, so all our experiments use same architecture for the generator.  In all our experiments, we are equipping the generator with conditional BN. This includes our experiments with "AC-GANs", as well as "concat" model and "projection" model.
> We can indeed explore the same result with generators equipped with a different way to introduce the conditional information (such as label concatenation); however, we intend not to make this (generator structure) the focus of our paper. On the base of our theoretical motivations, (Section 3) we also believe that our way will perform well even with a different way of label conditionalization of the generator.
> We would also like to emphasize that our projection model is prevailing on the super-resolution task as well, suggesting that our success is not the model and task specific.
> Lastly, as interesting as it is, we would not find a way to include the dependence of the performance on the image resolution into the scope of our paper.
>
>
> >2)
> >FID in real data. The numbers in Table 1 appear favorable to the projection model. Please add error bars (based on Figure 4, I would imagine they are quite large).
>
> We are sorry, but we are little confused about this suggestion. First of all, we are dealing with images from different classes in our experiments. The difficulty of image generation differs across each class, and the intra FID shall depend on the dataset of each class.  We, therefore, found no particular need for showing the size of its variance (error bar) in this experiment.  The goal of our Figure 4 here is to simply show that our projection method outperforms "concat" and "AC-GANs" on "most of the classes",  and we felt it more appropriate to visualize our claim with scatter plot.
>
>
> >Additionally, would it be possible to compute this statistic for *real* images? I would be curious to know what the FID looks like as a 'gold standard.'
>
> Please take a look at the definition of FID(p5, (Heusel et al., 2017)) .  FID is a measure of a difference between two distributions. If there are infinitely many 'real' images, the FID between 'real' images against 'real' images is trivially 0.  In our paper, we are comparing the empirical distribution of generated samples over 5000 samples against the that of the training 'real' images.  If we compute the empirical distribution of 'real' images against another empirical distribution of the 'real' images,  we are bound to observe some nonzero FID value.  However, we find no particular importance in computing such value.
>
>
> > Minor comments:
> >- I believe you have the incorrect reference for conditional batch normalization on Page 5.
> >- Please enlarge images in Figure 5-8. Hard to see the detail of 128x128 images.
> >- Please add citations for Figures 1a-1b. Do these correspond with some known models?
>
> Thanks for pointing out the incorrect references! We would revise the designated citations accordingly. We would also like to modify the figure images to improve the visuality.

---

### Official Review · AnonReviewer1 · 2017-11-25
**simple, interesting GAN modification; great results**

**Rating:** 7
**Confidence:** 5

**Review:**

The paper proposes a simple modification to conditional GANs, obtaining impressive results on both the quality and diversity of samples on ImageNet dataset. Instead of concatenating the condition vector y to the input image x or hidden layers of the discriminator D as in the literature, the authors propose to project the condition y onto a penultimate feature space V of D (by simply taking an inner product between y and V) . This implementation basically restricts the conditional distribution p(y|x) to be really simple and seems to be posing a good prior leading to great empirical results.

+ Quality:
- Simple method leading to great results on ImageNet!
- While the paper admittedly leaves theoretical work for future work, the paper would be much stronger if the authors could perform an ablation study to provide readers with more intuition on why this work. One experiment could be: sticking y to every hidden layer of D before the current projection layer, and removing these y's increasingly and seeing how performance changes.
- Appropriate comparison with existing conditional models: AC-GANs and PPGNs.
- Appropriate (extensive) metrics were used (Inception score/accuracy, MS-SSIM, FID)

+ Clarity:
- Should explicitly define p, q, r upfront before Equation 1 (or between Eq1 and Eq2).
- PPG should be PPGNs.

+ Originality:
This work proposes a simple method that is original compared existing GANs.

+ Significance:
While the contribution is significant, more experiments providing more intuition into why this projection works so well would make the paper much stronger.

Overall, I really enjoy reading this paper and recommend for acceptance!

---

> ### Author Response · Authors · 2017-12-03
> **Response to AnonReviewer1**
>
> We are very glad to hear that you enjoy our manuscript!
>
> > While the paper admittedly leaves theoretical work for future work, the paper would be much stronger if the authors could perform an ablation study to provide readers with more intuition on why this work. One experiment could be: sticking y to every hidden layer of D before the current projection layer, and removing these y's increasingly and seeing how performance changes.
> >While the contribution is significant, more experiments providing more intuition into why this projection works so well would make the paper much stronger.
>
> Ablation study was in fact a vexing issue in our paper, and we are still unsure of a way to theoretically back up our results. We may attempt your suggestion, and meanwhile continue looking for still other convincing experiments.
>
>
> > Should explicitly define p, q, r upfront before Equation 1 (or between Eq1 and Eq2).
> > PPG should be PPGNs.
>
> Thanks for pointing out the mistakes! We will make changes accordingly in the revised version.

---

### Official Review · AnonReviewer2 · 2017-11-27
**An unusually thorough GAN paper.**

**Rating:** 6
**Confidence:** 4

**Review:**

This manuscript makes the case for a particular parameterization of conditional GANs, specifically how to add conditioning information into the network.  It motivates the method by examining the form of the log density ratio in the continuous and discrete cases.

This paper's empirical work is quite strong, bringing to bare nearly all of the established tools we currently have for evaluating implicit image models (MS-SSIM, FID, Inception scores).

What bothers me is mostly that, while hyperparameters are stated (and thank you for that), they seem to be optimized for the candidate method rather than the baseline. In particular, Beta1 = 0 for the Adam momentum coefficient seems like a bold choice based on my experience. It would be an easier sell if hyperparameter search details were included and a separate hyperparameter search were conducted for the candidate and control, allowing the baseline to put its best foot forward.

The sentence containing "assume that the network model can be shared" had me puzzled for a few minutes. I think what is meant here is just that we can parameterize the log density ratio directly (including some terms that belong to the data distribution to which we do not have explicit access). This could be clearer.

---

> ### Author Response · Authors · 2017-12-03
> **Response to AnonReviewer2**
>
>
> >What bothers me is mostly that, while hyperparameters are stated (and thank you for that), they seem to be optimized for the candidate method rather than the baseline. In particular, Beta1 = 0 for the Adam momentum coefficient seems like a bold choice based on my experience.
>
> We did not perform any hyper-parameter optimization for the Adam optimizer and the number of critic updates, etc…
> We just used the same hyper-parameters used in Gulrajani et al. (2017, https://github.com/igul222/improved_wgan_training/blob/master/gan_cifar_resnet.py ), because we adopted the practically the same architecture used in the very paper.
> We must admit that we simply could not spare enough time for the parameter search for the ImageNet experiments.  However, we plan to do the search for (beta1, alpha of Adam) on CIFAR 10 or CIFAR 100 and compare the performance against "AC-GANs", "concat" and "projection".
>
>
> >The sentence containing "assume that the network model can be shared" had me puzzled for a few minutes. I think what is meant here is just that we can parameterize the log density ratio directly (including some terms that belong to the data distribution to which we do not have explicit access). This could be clearer.
>
> Thank you very much! We concur with you in your views and we will reflect the suggestion on this part of the revision.

---

> > ### Author Response · Authors · 2017-12-19
> > **Reference**
> >
> > We are sorry but we forgot to note the reference information of "Gulrajani et al. (2017)" in the previous comment.
> >
> > Reference:
> > Improved training of Wasserstein GANs
> > Ishaan Gulrajani, Faruk Ahmed, Martin Arjovsky, Vincent Dumoulin and Aaron Courville
> > In NIPS2017

---

> ### Author Response · Authors · 2017-12-19
> **We conducted hyper parameter search on CIFAR-100 dataset, and confirmed our projection model achieved better performance on all choice of the hyper-parameters.**
>
> We reflected your suggestion on our revision and conducted the hyper-parameter search on CIFAR-100 about the Adam hyper-parameters (learning rate $\alpha$ and 1st order momentum $\beta_1$).
> Namely, we varied each one of these parameters while keeping the other constant, and reported the inception scores for all methods including several versions of “concat” architectures to compare.
> More specifically, we tested with concatenation module introduced at (a) input layer, (b) hidden layer, and at (c) output layer.  The results of this complementary experiment are now provided in the of the appendix section A of the revised paper.
>
> As we can see in Figure 11, our “projection” architecture excelled over all other architectures for all choice of the hyper-parameters, and achieved the inception score of 9.53.  Meanwhile,  concat architectures were able to achieve all 8.82 at most.
> The best concat model in term of the inception score on CIFAR-100 was the hidden concat with $\alpha$=0.0002 and $\beta_1$ = 0, which turns out to be the very choice of the parameters we picked for the original ImageNet experiment. Unfortunately,  we were not able to secure the time for the parameter search on ImageNet experiment. However, from the way the outcomes look for the CIFAR-100, we speculate the same to happen.

---

### Public Comment · ~Florian_Strub1 · 2017-11-28
**Confusion in References**

In the paper, the authors use Conditional Batch Normalization and refer to the following paper:
Vincent Dumoulin, Ishmael Belghazi, Ben Poole, Alex Lamb, Martin Arjovsky, Olivier Mastropietro, and Aaron Courville. Adversarially learned inference. In ICLR, 2017.
Although this paper is related to adversarial learning, it is not related to Conditional Batch Normalization.

I believe there may be some confusion in the references. The following papers may be more relevant as they both introduce Conditional Normalization in different contexts:
Dumoulin, V., Shlens, J., and Kudlur, M. A learned representation for artistic style. In ICLR, 2017.
de Vries, H., Strub, F., Mary, J., Larochelle, H., Pietquin, O., and Courville, A. Modulating early visual processing by language. In NIPS, 2017.

Interestingly, subsequent work has shown that the effect of this form of conditioning can be decorrelated from normalization layers, thus referring to the method as Feature-wise Linear Modulation, or FiLM:
Perez E., Strub F., de Vries H., Dumoulin V., Courville A. FiLM: Visual Reasoning with a General Conditioning Layer. In AAAI, 2018.

It may also be worthwhile to consider updating the name used in the paper from Conditional Batch Normalization to FiLM, to follow the latest literature on this method.

---

> ### Author Response · Authors · 2017-11-29
> **Thanks for pointing out the mistake!**
>
> We were clearly making typos in the reference.
> The reference you mentioned is the very reference we intended to cite.
> As for the use of the word  “FiLM”, we would like to stick for now to the “conditional batch normalization” to make it easy for the readers to readily catch the framework of our algorithm.

---

### Author Response · Authors · 2017-12-03
**Thank you so much for the reviews!**

We thank all three reviewers for thorough reading of our manuscript and their comments and suggestions.
We responded to all the suggestions and made corrections for each reviewer’s comment separately.

---

### Author Response · Authors · 2017-12-19
**Uploaded the revision**

We owe great thanks to all reviewers for helpful comments to improve our manuscripts.
We revised our manuscript based on the reviewer’s comments and uploaded the revision.
For an important note, we re-calculated the inception scores and FID with the original evaluation code written in TensorFlow,  because the results slightly differed from our rendition written in Chainer.
Please rest assured, however, because the newly computed results does not affect any claims we have made on the original version of our paper.

Also, to show the efficacy of our method on smaller benchmark datasets,  we added the results on CIFAR-10 and CIFAR-100 datasets.
Our projection model was able to eclipse the comparative models (concat discriminator models and AC-GANs) on these datasets as well.
Please see Appendix A for details.

---

### Author Response · Authors · 2018-02-02
**The code for reproducing the results**

The code for reproducing the results in this paper has been uploaded at
https://github.com/pfnet-research/sngan_projection.
Also we have uploaded other materials (pretrainied models, generated images and movies) at https://drive.google.com/drive/folders/1GnDuF02F3a_zNEwiA74DnaG7OQ3-Co3N.
Please go to the links if you are interested in our work.

---

### Decision · Program_Chairs · 2018-01-29
**ICLR 2018 Conference Acceptance Decision**

**Decision:**

Accept (Poster)

**Comment:**

The paper proposes a simple modification to conditional GANs, where the discriminator involves an inner product term between the condition vector y and the feature vector of x. This formulation is reasonable and well motivated from popular models (e.g., log-linear, Gaussians). Experimentally, the proposed method is evaluated on conditional image generation and super-resolution tasks, demonstrating improved qualitative and qualitative performance over the existing state-of-the-art (AC-GAN).